# Practical suggestions for harms reporting in exercise oncology: the Exercise Harms Reporting Method (ExHaRM)

Rosalind R Spence [ID],[1,2] Carolina X Sandler [ID],[2,3] Tamara L Jones,[2,4] Nicole McDonald,[1,2] Riley M Dunn,[2] Sandra C Hayes [ID] [1,2,5]

[1]Menzies Institute of Health Queensland, Griffith University, Brisbane, Queensland, Australia
[2]Improving Health Outcomes for People (ihop) research group, Brisbane, Queensland, Australia
[3]Sport and Exercise Science, School of Health Science, Western Sydney University, Sydney, New South Wales, Australia
[4]Melbourne School of Psychological Sciences, Faculty of Medicine, Dentistry, and Health Sciences, University of Melbourne, Melbourne, Victoria, Australia
[5]School of Health Sciences and Social Work, Griffith University, Brisbane, Queensland, Australia

**Correspondence to**
Dr Rosalind R Spence;
r.spence@griffith.edu.au

## ABSTRACT

The volume of high-quality evidence supporting exercise as beneficial to cancer survivors has grown exponentially; however, the potential harms of exercise remain understudied. Consequently, the trade-off between desirable and undesirable outcomes of engaging in exercise remains unclear to clinicians and people with cancer. Practical guidance on collecting and reporting harms in exercise oncology is lacking. We present a harms reporting protocol developed and refined through exercise oncology trials since 2015.

Development of the Exercise Harms Reporting Method (ExHaRM) was informed by national and international guidelines for harms reporting in clinical trials involving therapeutic goods or medical devices, with adaptations to enhance applicability to exercise. The protocol has been adjusted via an iterative process of implementation and adjustment through use in multiple exercise oncology trials involving varied cancer diagnoses (types: breast, brain, gynaecological; stages at diagnosis I–IV; primary/recurrent), and heterogeneous exercise intervention characteristics (face to face/telehealth delivery; supervised/unsupervised exercise). It has also involved the development of terms (such as, adverse outcomes, which capture all undesirable physical, psychological, social and economic outcomes) that facilitate the harms assessment process in exercise.

ExHaRM involves: step 1: Monitor occurrence of adverse outcomes through systematic and non-systematic surveillance; step 2: Assess and record adverse outcomes, including severity, causality, impact on intervention and type; step 3: Review of causality by harms panel (and revise as necessary); and step 4: Analyse and report frequencies, rates and clinically meaningful details of all-cause and exercise-related adverse outcomes.

ExHaRM provides guidance to improve the quality of harms assessment and reporting immediately, while concurrently providing a framework for future refinement. Future directions include, but are not limited to, standardising exercise-specific nomenclature and methods of assessing causality.

## INTRODUCTION

Since publication of the first exercise oncology prescription guidelines over a decade ago,[1 2] the volume and quality of evidence supporting the benefits of exercise following a cancer diagnosis has increased exponentially.[3 4] The current model of evidence-based medicine is founded on the principle that expected benefits, as well as potential harms, patient values and preferences, drive clinician recommendations and patient decisions.[5] Unfortunately, the harms of exercise remain understudied and under-reported in exercise trials (missing or incompletely reported in 51%–75% of studies),[6 7] and specifically, within exercise oncology (not reported in 34%–47% of studies).[8 9] Harms are 'the totality of possible adverse consequences of an intervention or therapy; they are the direct opposite of benefits, against which they must be compared.'[10] The absence of data regarding the harms of exercise significantly compromises the ability of clinicians, patients and exercise professionals to evaluate the trade-off between desirable and undesirable consequences.[5] There is insufficient evidence for (1) clinicians to judge the safety profile of exercise as a potential adjuvant therapy, (2) patients to make informed decisions about potential consequences of exercise participation and (3) exercise professionals to determine the appropriateness of exercise (ie, contraindications, or dosage and prescription considerations).

Despite a growing recognition of the importance of accurate and comprehensive reporting of the harms of exercise,[7] practical guidance on how to manage the collection and reporting process in exercise oncology interventions is missing.[8 9] Well-accepted standards and guidelines for harms reporting in pharmacological cancer clinical trials exist.[11] However, the direct application of these frameworks to exercise trials, without modification, is inappropriate.[12] Pharmacological harms are assessed via adverse events (AEs) (ie, 'any undesirable medical occurrence'.[13] However, the collection of only AEs fails to capture the breadth of potential negative

BMJ

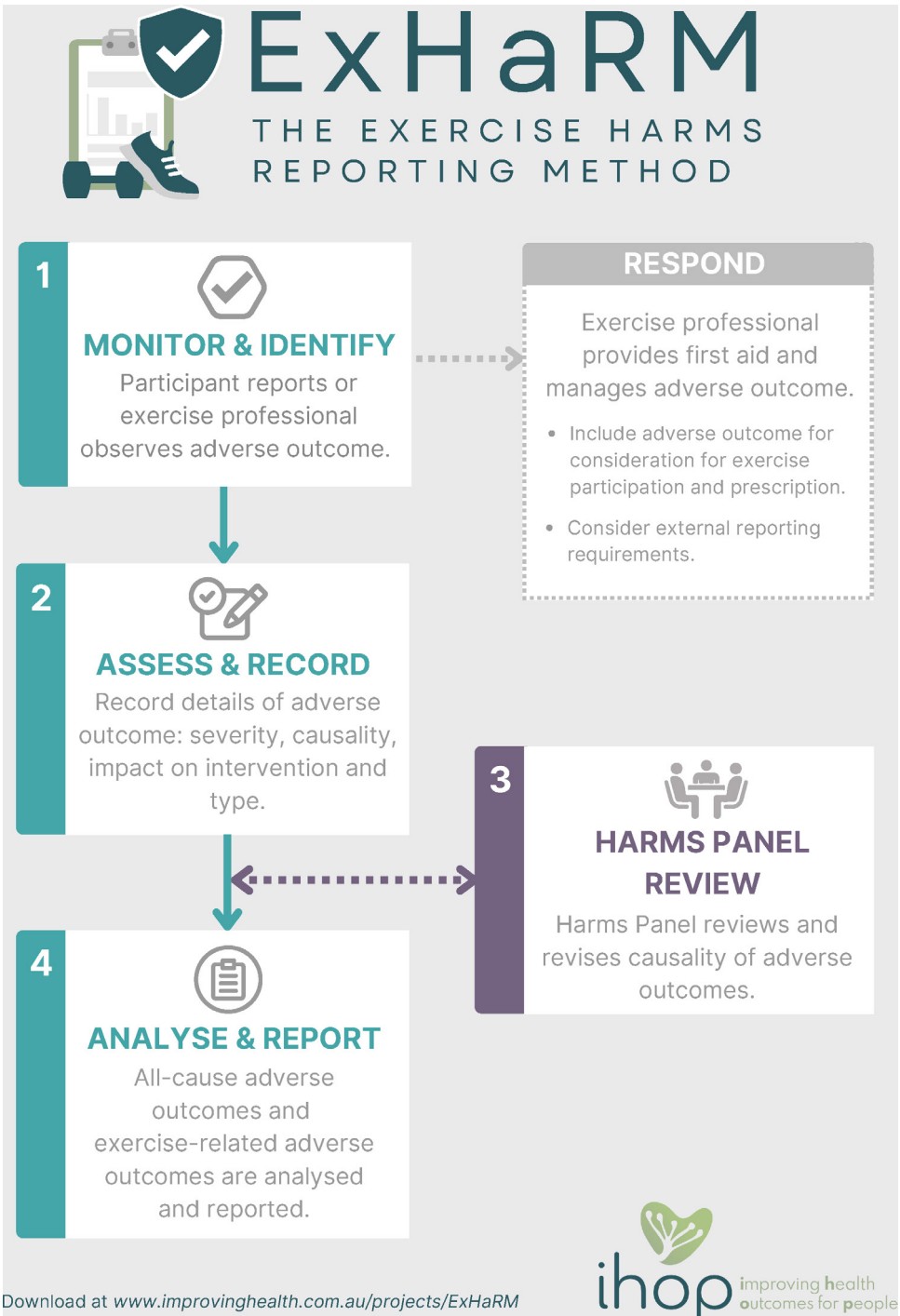

**Figure 1** An overview of the Exercise Harms Reporting Method (ExHaRM). Note: See online supplemental material box 1 for definitions and further explanation of terms used in this flow chart.

consequences that may occur due to behavioural interventions, including exercise.[12] Harms reporting in exercise interventions is further complicated by the need to distinguish harms associated with medical treatment from those associated with exercise. Below we share our approach to exercise harms reporting, with the aim of encouraging more comprehensive reporting of harms data and providing a platform and framework from which future studies can build.

## EXERCISE HARMS REPORTING METHOD

The Exercise Harms Reporting Method (ExHaRM) is a protocol for the monitoring and reporting of exercise-related harms. It was developed based on guidelines for harms reporting in clinical trials involving therapeutic goods or medical devices (specifically, Good Clinical Practice guidelines[13]; National Cancer Institute Common terminology criteria for adverse events[11]), as well as international,[14] Australian and local institutional reporting requirements,[15]

and the expert opinions of trial investigators (including exercise oncology researchers, accredited exercise physiologists, cancer nurses, breast surgeons, gynaecological oncologists and medical oncologists). Our team first implemented an intentional harm assessment and reporting method in 2015, with the commencement of exercise oncology trials targeting cancer survivors with multiple comorbidities[16] or undergoing intense treatment,[17] who were potentially higher-risk exercisers than those included in previous trials. Since that time, the protocol has been refined through an iterative process of implementation and adjustment. Iterations of ExHaRM have been used in four exercise oncology trials,[16–19] involving samples with breast, brain and gynaecological cancer, different stages of disease at diagnosis (stages I–IV), primary and recurrent diagnoses, and the presence of multiple comorbidities. Further, ExHaRM has been used in studies that involve mixed-mode exercise (aerobic and resistance training) delivered during and following treatment, via various modes of delivery (face-to-face supervision ranging from two sessions/week to one session/month, in addition to trials delivered via telehealth), and a range of settings (eg, exercise completed at participants' homes, community fitness centres and university exercise clinics).

ExHaRM defines the harms of exercise as all undesirable physical, psychological, economic or social consequences (covering incidences, experiences, occurrences) that are related to an individual's participation in exercise. Given assessment of adverse events does not fully capture harms of exercise, we, therefore, propose the use of the term adverse outcomes to assess undesirable physical, psychological, social and economic outcomes, experiences or events, irrespective of whether they have a causal relationship with exercise. ExHaRM involves four core steps (summarised in figure 1 and explained below) to capture, classify, analyse and report all-cause and exercise-related adverse outcomes. A downloadable and interactive version of ExHaRM is available at www.improvinghealth.com.au/projects/exharm). Key terms, including their definitions and related information (eg, prompts or questions to ensure accurate data collection, and coding recommendations to aid efficient recording) are denoted by italics throughout the method and can be in online supplemental material box 1.

### Details of the reporting protocol

The purpose of the The Exercise Harms Reporting Method (ExHaRM) is to guide the collection and reporting of informative data on the *harms of exercise*.

Note: Italicised terms are defined in online supplemental material box 1.

### Step 1: Monitor and Identify

Systematically monitor and identify *adverse outcomes*:

▶ Actively prompt participants to report *adverse outcomes*. Use standardised language and standardised *recall periods*. For example, the exercise professional asks the participant at every contact: 'Since our last session have you experienced any harmful or undesirable outcomes, whether or not you think they were caused by exercise? For example, have you had an injury or worsening of an old injury, or any unusual or worsening treatment-related symptom?' (ie, *active surveillance*).

▶ Train exercise professionals to actively observe for *adverse outcomes* that may occur during supervised exercise, and to probe during discussions if a participant reports something that might be an adverse outcome (ie, *passive surveillance*/spontaneous reporting of *adverse outcomes*).

▶ Consider generating a study-specific list of *adverse outcomes of interest* that are systematically recorded via a checklist or standardised test at fixed time points (*systematic surveillance*), in addition to the broad question recommended above (*non-systematic surveillance*).

### Respond

While not part of the harms-reporting process, the exercise professional has a duty of care to the participant to identify, consider and manage all *adverse outcomes* and depending on the nature of the *adverse outcome* (with particular attention required for *adverse events*) may have external reporting requirements:

▶ Provide first aid and management of the *adverse outcome*: Respond to immediate needs of the participant through administration of first aid, cessation or modification of exercise, referral to other health professional and/or other responses as required.

▶ Complete tasks as per exercise professional duties and scope of practice: Consider whether *adverse outcome* requires an absence from exercise (ie, is it a contraindication?) and/or modification of exercise parameters (eg, volume, intensity or type).

▶ Report as per requirements of employer, institution, funding and/or ethics committees.

Note: The definitions (eg, of *harms*, *adverse outcomes* and *severity*) in this protocol are those determined to be most appropriate to the evaluation of exercise-related harms. Regulatory committees (eg, a human research ethics committee) typically require the reporting of '*serious adverse events*' only, which will be captured within the monitoring and reporting of *adverse outcomes* as outlined in this protocol.

### Step 2: Assess and Record

Assess and record the *adverse outcome* and associated classifications of *severity*, *causality*, *impact of adverse outcome on the intervention* and *type of adverse outcome*, including sufficient detail to justify the assigned category. Also record detail of the context and sequelae of the *adverse outcome*, for example, where and when the *adverse outcome* occurred, what clinical action was taken, the duration of the adverse outcome and any subsequent treatment.

▶ *Severity (Grades 1–5)*: based primarily on the resultant medical treatment and the impact of the *adverse outcome* on activities of daily living.

▶ *Causality*: based on the timing of the *adverse outcome* in relation to exercise and plausibility of

relationship with exercise or other factors (see useful questions to consider under 'causality'). Include both the participant's opinion and exercise professional's judgement of causality.

▶ *Impact on intervention*: based on changes to a participant's involvement in the exercise intervention caused by the *adverse outcome*. This may involve modifications (eg, modified exercise prescription, missed exercise sessions) or absences from the intervention (either temporary or permanent).

▶ *Type of adverse outcome*: the purpose of categorising *adverse outcome* by type is to create groupings for simplified summary based on predetermined categories (eg, abnormal response to exercise, injury, exacerbation of treatment-related side effect).

Note: Care should be taken to differentiate between a *persistent* and *recurrent adverse outcome*, as *persistent adverse outcomes* need only be reported once, whereas *recurrent adverse outcomes* should be reported at each recurrence.

### Step 3: Review by *harms panel* and potential revision of causality

The attribution of *causality* (ie, determining which *adverse outcomes* are causally related to exercise) is essential to the evaluation of exercise-related harms. *Causality* attribution requires knowledge of the mechanisms by which an intervention may cause harm, as well as the expected responses to an intervention (ie, scope of exercise professional). Concurrently, an understanding of what is common and expected in the population (both disease and treatment related) is also required and is likely most appropriately provided by the medical team. Therefore, while it is appropriate for exercise professionals to make the initial judgement of causality, ideally adverse outcomes should be reviewed by a panel with additional expertise, and preferably at least one individual who is not directly involved in the study.

All *adverse outcomes* are reported to the study *harms panel* on a predetermined schedule. This may be frequently (eg, monthly) or at a single point prior to data analysis.

The *harms panel* review all *adverse outcomes* with special attention to:

▶ Unusual or unanticipated *adverse outcomes* (as judged by panel).

▶ Discordant opinions of causality recorded between participant and exercise professional.

▶ *Adverse outcomes* ≥grade 3.

▶ *Adverse outcomes* that the exercise professional has highlighted for review.

Discrepancies in *causality* attribution between exercise professional and any member of the harms panel should be discussed, and the consensus reached by the panel be recorded as the final *causality* decision.

The *harms panel* may also determine that a reported *adverse outcome* does not meet the definition of an *adverse outcome* (eg, the recorded event might not be judged by the panel as being undesirable). In this situation, the *adverse outcome* would be removed from the dataset and would not be reported.

### Step 4: Analyse and report
#### *Data preparation and analysis*

▶ Any *adverse outcome* with a *causality* of certain, likely or possible is categorised as an *exercise-related adverse outcome*. The term '*all-cause adverse outcomes*' is then used to refer to *adverse outcome* of any *causality* (ie, all *adverse outcomes*: those deemed exercise related, as well as those not attributed to exercise).

▶ Recode *adverse outcomes* with standardised language: each *adverse outcome* identified in the study should be described using standardised language (eg, *Medical dictionary for Regulatory Activities* (MedDRA)[20] preferred terms) and summarised (eg, within system organ class or appropriate groupings). For example: a report of 'stiff and sore shoulders' as an *adverse outcome* in the clinical notes could be coded as 'arthralgia' using *MedDRA* terminology and listed under the category of 'Musculoskeletal and Connective Tissue Disorders', along with other conditions, such as arthritis.

▶ Group all-cause *adverse outcomes* and *exercise-related adverse outcomes* by each category of type of adverse outcome and *impact on intervention*.

#### *Reporting*
*All-cause* and *exercise-related adverse outcomes*:

▶ Report the number of *all-cause adverse outcomes* and *exercise-related adverse outcomes*, including the number (and proportion) of *all-cause adverse outcomes* and *exercise-related adverse outcomes* categorised by *severity* as mild-moderate (grades 1 and 2) and severe (grades 3–5).

▶ For these harms outcomes (ie, number of *all-cause adverse outcomes* and *exercise-related adverse outcomes* by severity categories) it is helpful to also report the number of participants (and proportion of all participants) reporting an *adverse outcome* at least once.

▶ The number of *adverse outcomes* (and percentage) for each MedDRA code (eg, arthralgia) and system organ class (eg, musculoskeletal and connective tissue disorders), or other standardised language groupings, should be reported for *all-cause adverse outcomes* and for *exercise-related adverse outcomes*.

Exercise-related adverse outcomes only:

▶ The number (and proportion) of *exercise-related adverse outcomes* in each category of '*impact on intervention*' and '*type of adverse outcome*' should be reported.

▶ Highlight any unique or key *exercise-related adverse outcomes* relevant to the context of the study, or of clinical relevance (eg, lymphoedema, dislodged peripherally inserted central catheter, bone fractures).

▶ Consider reporting *exercise-related adverse outcomes* as a rate per person per week of intervention, in conjunction with the average weekly minutes of exercise that were completed. If rate is not reported, then ensure sufficient intervention and trial details are included to allow subsequent studies to compare results.

► *Adverse outcomes* that occur during exercise testing (ie, objective data collection) should be reported separately to those that occur during the intervention.

## Considerations and future directions

Exercise is frequently reported as being safe for those diagnosed with cancer. Yet quality data supporting this statement is lacking. Our understanding of the need to collect harms data, and subsequently, optimal methods for assessing and reporting harms has evolved. For a decade before the first iteration of ExHaRM, our trials involved intentional collection of disease-related and treatment-related side effects, as well as undesirable responses to exercise, for the purpose of exercise prescription modification. With the advent of trials including participants for whom exercise was potentially more likely to cause undesirable outcomes, and exploring less controlled exercise settings, recording and reporting safety-related data became a higher priority. Through these trials, ExHaRM progressed to include systematic recording and reporting of adverse events, the assessment of additional relevant information (eg, causality, severity, impact) and then exercise-specific nomenclature and definitions (eg, adverse outcomes, as defined in the current version).

ExHaRM now presents an exercise harms reporting protocol ready for widespread use, providing practical, exercise-specific guidelines for the collection and reporting of harms in exercise oncology research. In line with best practice,[21] ExHaRM facilitates reporting of any, and all, adverse outcomes, of any severity and causality. In doing so, ExHaRM supports collection of exercise-related adverse outcomes required for quantifying harms of exercise, while concurrently (1) collects information necessary for targeted, individualised exercise prescription; (2) reduces the influence of potential participant or exercise professional bias on harms reporting; (3) allows for exploring the rate of exercise-related adverse outcomes to all-cause adverse outcomes, which may help identify attribution bias[21] and (4) provides the necessary data for inclusion in future meta-analyses to determine the strength of harms evidence in exercise oncology and to evaluate whether specific patient (eg, age, cancer type, stage of disease at diagnosis) or exercise characteristics (eg, frequency, intensity, mode, mode of delivery) influence harms.

We recognise that just as is the case for assessing potential exercise benefit, the goal for future ExHaRM iterations will be to reduce data collection burden (to participants, researchers, funding bodies, etc) while maintaining comprehensive collection of meaningful information about exercise harms. Integrating a checklist of expected and clinically-important exercise-related harms (eg, similar to the Common Terminology Criteria for Adverse Events[11] and the associated patient-reported version[22]) is a proposed development for future iterations of ExHaRM. However, while this may prove useful in streamlining harms data collection, it would not replace the need to collect adverse outcomes through more time-consuming non-systematic approaches as promoted in the current ExHaRM version.

The identification of those adverse outcomes that are exercise-related is essential to the evaluation of the harms of exercise. The validity of the causality assessment process described in ExHaRM has not been formally evaluated and relies on experienced exercise and medical clinicians being involved, at a minimum, in the reviewing of causality attribution. Development of an algorithm similar to the causality probability scale developed by Naranjo *et al*[23] would be one approach to standardising causality attribution in exercise oncology. However, in the interim, ExHaRM provides clear guidance to exercise professionals in judging the exercise-relatedness of adverse outcomes. We have found the involvement of a harms panel in ExHaRM extremely useful in reducing the potential for over-reporting and under-reporting exercise-related adverse outcomes and ensuring consistency in the classification of adverse outcomes.

ExHaRM does not include a threshold at which exercise would be defined safe or unsafe. Decisions regarding the role of exercise in a specific context should be based on weighing evidence of the expected benefits against the potential harms. As is true with medical treatments, in some situations the benefits are sufficient that a higher level of concurrent harm is deemed acceptable, such as may be the case for life-saving chemotherapy. Within exercise oncology, some patient groups (eg, those with advanced bone lesions or metastases) may be at higher risk of harm from exercise; however, if the benefits were of sufficient value to the population (eg, maintained ability to independently complete activities of daily living) then exercise may be incorporated among care despite the harm profile.

With iterations of ExHaRM there was increasing recognition of the need to identify and consistently apply appropriate terms. Much discussion and deliberation has occurred over the use of all harms-related terms, but in particular the use of harms vs safety (which presupposes a positive outcome), and adverse outcomes versus adverse events. These discussions acknowledged that there are unintended negative consequences (ie, harms) of changing lifestyle behaviours that do not meet the definition of an adverse event as used in pharmacological harms reporting.[12] Adverse outcome (as used in the current iteration of ExHaRM) is a widely encompassing term which addresses the limitations of the term adverse event. There was also deliberation of whether adverse outcomes classified with the causality category of 'possible' should be defined, along with those specified as 'certain' and 'likely', as exercise-related adverse outcomes. This issue of causality categorisation also remains an area of debate in the harms reporting of pharmacological clinical trials.[24] Our current approach to include 'possible' is more inclusive, which was considered appropriate given the infancy of

this specific area of harms reporting, but is a topic for future discussion and debate.

ExHaRM was developed for use in research. However, with minimal modifications this protocol could also be implemented in hospital-based, university-based and community-based exercise settings. Monitoring adverse outcomes (and near misses, in which no adverse outcome occurred but could have) within a clinic would facilitate risk mitigation through modification of processes (e.g., modification of prescription, type of equipment used, inclusion of contraindications or conditions that should be prescreened), with the goal of reducing harms without compromising benefit. The collection of harms data in real-world settings also allows for the identification of rare adverse outcomes not previously identified during research, as well as those that may only occur outside of controlled research settings. Further, this harms reporting framework may also prove useful in exercise trials outside the oncological setting, as well as other lifestyle intervention trials beyond exercise.

## SUMMARY

There is a clear imperative to improve harms reporting in exercise oncology, and this imperative is heightened with the inclusion of understudied, rare cancers and potentially higher-risk cohorts and interventions (eg, participants with poorer prognoses and/or extensive comorbidities; higher intensity and/or less supervised exercise interventions) in exercise trials. In the short term, as per Consolidated Standards of Reporting Trials (CONSORT) guidelines,[10 25] all exercise trials should have a harms reporting protocol. In the longer term, consensus regarding a standardised protocol and consistent use of harms-related terms, such as adverse outcomes, will facilitate rapid progress in this specific area of exercise oncology research.

**Acknowledgements** The authors would like to acknowledge the contribution of participants, exercise professionals and chief investigators who were part of the SAFE, BRACE, ECHO and ECHO-R trials that were fundamental in the development of ExHaRM.

**Contributors** RRS: conceptualisation, methodology, investigation, data curation, writing—original draft. CXS: investigation, data curation, writing—review and editing. TLJ: data curation, writing—review and editing. NM: visualisation, writing—review and editing. RMD: investigation, writing—review and editing. SCH: conceptualisation, supervision, data curation, writing—review and editing, funding acquisition.

**Funding** The suite of ihop exercise oncology trials that have contributed to the work presented have received funding from the following bodies: Icon Cancer Foundation (BRACE trial), Medical Research Future Fund (ECHO-R trial) and Cancer Australia, Cancer Council Australia, World Cancer Research Fund, Cancer Council Queensland and Griffith University (ECHO trial). CS is supported by a Cancer Institute New South Wales Early Career Fellowships (2021/ECF1310).

**Competing interests** None declared.

**Provenance and peer review** Not commissioned; externally peer reviewed.

peer-reviewed. Any opinions or recommendations discussed are solely those of the author(s) and are not endorsed by BMJ. BMJ disclaims all liability and responsibility arising from any reliance placed on the content. Where the content includes any translated material, BMJ does not warrant the accuracy and reliability of the translations (including but not limited to local regulations, clinical guidelines, terminology, drug names and drug dosages), and is not responsible for any error and/or omissions arising from translation and adaptation or otherwise.

**ORCID iDs**
Rosalind R Spence http://orcid.org/0000-0002-5446-5562
Carolina X Sandler http://orcid.org/0000-0001-9826-2487
Sandra C Hayes http://orcid.org/0000-0002-7005-5184

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
