## [Reviewer comments · BMJ Open]

ARTICLE DETAILS

TITLE (PROVISIONAL)	Practical suggestions for harms reporting in exercise oncology: the Exercise Harms Reporting Method (ExHaRM)
AUTHORS	Spence, Rosalind; Sandler, Carolina; Jones, Tamara; McDonald, Nicole; Dunn, Riley; Hayes, Sandra

VERSION 1 – REVIEW

REVIEWER	Schwartz, Anna L.
REVIEW RETURNED	18-Oct-2022

GENERAL COMMENTS	This is an outstanding paper that is well written and provides a clear explanation of a solid protocol to monitor and report exercise-related harms in exercise oncology clinical trials. The protocol has gone through an iterative process of development and refinement to propose different types of adverse outcomes related to exercise. The ExHaRM reporting method is clearly described in Table 1. The authors have provided clear limits of the use of the tool and declared that further research is needed prior to clinical application. ExHaRM has the potential to be a significant contribution to the future of how research and the practice of exercise oncology is conducted. This work provides a strong contribution to exercise oncology.
---

REVIEWER	Neil-Sztramko, Sarah McMaster University
REVIEW RETURNED	28-Oct-2022

GENERAL COMMENTS	This communication presents ExHARM, a tool developed by the authors to foster rigorous tracking and reporting of harms in exercise oncology research. The authors have strong justification for creating such a tool given the challenges outlined in using other tools that were designed and are better suited for pharmacological studies, as well as the documented limited reporting of adverse events in this literature. An impressive amount of work has gone into this, and I commend the authors for sharing it with others. As part of this communication, I would like to hear more about how the tool was developed, perhaps how it has evolved, and how it has been utilized. This would better allow the reader to judge its suitability for their context.
--

VERSION 1 – AUTHOR RESPONSE

Reviewer: 1

Anna L. Schwartz

Comments to the Author:

This is an outstanding paper that is well written and provides a clear explanation of a solid protocol to monitor and report exercise-related harms in exercise oncology clinical trials. The protocol has gone through an iterative process of development and refinement to propose different types of adverse outcomes related to exercise. The ExHaRM reporting method is clearly described in Table 1. The authors have provided clear limits of the use of the tool and declared that further research is needed prior to clinical application. ExHaRM has the potential to be a significant contribution to the future of how research and the practice of exercise oncology is conducted. This work provides a strong contribution to exercise oncology.

Response:

We thank the reviewer for their comments and encouragement and the time given to review this submission.

** **

Reviewer: 2

Dr. Sarah Neil-Sztramko, McMaster University

Comments to the Author:

This communication presents ExHARM, a tool developed by the authors to foster rigorous tracking and reporting of harms in exercise oncology research. The authors have strong justification for creating such a tool given the challenges outlined in using other tools that were designed and are better suited for pharmacological studies, as well as the documented limited reporting of adverse events in this literature.

An impressive amount of work has gone into this, and I commend the authors for sharing it with others. As part of this Communication, I would like to hear more about how the tool was developed, perhaps how it has evolved, and how it has been utilized. This would better allow the reader to judge its suitability for their context.

Response:

Thank you for your comments and words of support. We hope that the additions throughout the manuscript address the reviewer's suggestions and provide additional insight for the reader. Two key changes are outlined below:

- a) We have included additional history of the development of the model to the methods paragraph, which also describes the types of cancer cohorts and exercise contexts the ExHaRM protocol has been used to evaluate.

The **Exercise Harms Reporting Method (ExHaRM)** is a protocol for the monitoring and reporting of exercise-related harms. It was developed based on guidelines for harms

reporting in clinical trials involving therapeutic goods or medical devices (specifically, Good Clinical Practice guidelines;[13] National Cancer Institute Common terminology criteria for AEs [CTC-AE][11]), as well as international,[14] Australian and local institutional reporting requirements,[15] and the expert opinions of trial investigators (including exercise oncology researchers, accredited exercise physiologists, cancer nurses, breast surgeons, gynaecological oncologists and medical oncologists). Our team first implemented an intentional harm assessment and reporting method in 2015, with the commencement of exercise oncology trials targeting cancer survivors with multiple comorbidities[16] or undergoing intense treatment,[17] who were potentially higher-risk exercisers than those included in previous trials. Since that time, the protocol has been refined through an iterative process of implementation and adjustment. Iterations of ExHaRM have been used in four exercise oncology trials,[16-19] involving samples with breast, brain and gynaecological cancer, different stages of disease at diagnosis (stage I-IV), primary and recurrent diagnoses, and the presence of multiple comorbidities. Further, ExHaRM has been used in studies that involve mixed-mode exercise (aerobic and resistance training) delivered during and following treatment, via various modes of delivery (face-to-face supervision ranging from 2 sessions/week to 1 session/month, in addition to trials delivered via tele-health), and a range of settings (e.g., exercise completed at participants' homes, community fitness centres, and university exercise clinics).

b) Additional content relating to the evolution of the method has been included in the discussion:

Our understanding of the need to collect harms data, and subsequently, optimal methods for assessing and reporting harms has evolved. For a decade before the first iteration of ExHaRM, our trials involved intentional collection of disease- and treatment-related side effects, as well as undesirable responses to exercise, for the purpose of exercise prescription modification. With the advent of trials including participants for whom exercise was potentially more likely to cause undesirable outcomes, and exploring less controlled exercise settings, recording and reporting safety-related data became a higher priority. Through these trials, ExHaRM progressed to include systematic recording and reporting of adverse events, the assessment of additional relevant information (e.g., causality, severity, impact) and then exercise-specific nomenclature and definitions (e.g., adverse outcomes, as defined in the current version).

VERSION 2 – REVIEW

REVIEWER	Neil-Sztramko, Sarah McMaster University
REVIEW RETURNED	10-Nov-2022
GENERAL COMMENTS	Thank you for this additional information, I will be happy to see this paper published and look forward to the contributions it will make to the exercise oncology literature.